# The Chemistry and Applications of Metal–Organic Frameworks (MOFs) as Industrial Enzyme Immobilization Systems

**DOI:** 10.3390/molecules27144529

**Published:** 2022-07-15

**Authors:** Allison R. M. Silva, Jeferson Y. N. H. Alexandre, José E. S. Souza, José G. Lima Neto, Paulo G. de Sousa Júnior, Maria V. P. Rocha, José C. S. dos Santos

**Affiliations:** 1Departamento de Engenharia Química, Campus do Pici, Universidade Federal do Ceará, Bloco 709, Fortaleza 60455760, Brazil; allisonruanms@gmail.com (A.R.M.S.); jeferson.yves@gmail.com (J.Y.N.H.A.); erick@aluno.unilab.edu.br (J.E.S.S.); zegadelha@hotmail.com (J.G.L.N.); 2Centro de Ciências, Departamento de Química Orgânica e Inorgânica, Campus do Pici, Universidade Federal do Ceará, Fortaleza 60455970, Brazil; paulogdsj@gmail.com; 3Instituto de Engenharias e Desenvolvimento Sustentável, Universidade da Integração Internacional da Lusofonia Afro-Brasileira, Campus das Auroras, Redencao CEP 62790970, Brazil

**Keywords:** metal–organic frameworks, enzymatic immobilization, enzymes, enzymatic catalysis, industrial application

## Abstract

Enzymatic biocatalysis is a sustainable technology. Enzymes are versatile and highly efficient biocatalysts, and have been widely employed due to their biodegradable nature. However, because the three-dimensional structure of these enzymes is predominantly maintained by weaker non-covalent interactions, external conditions, such as temperature and pH variations, as well as the presence of chemical compounds, can modify or even neutralize their biological activity. The enablement of this category of processes is the result of the several advances in the areas of molecular biology and biotechnology achieved over the past two decades. In this scenario, metal–organic frameworks (MOFs) are highlighted as efficient supports for enzyme immobilization. They can be used to ‘house’ a specific enzyme, providing it with protection from environmental influences. This review discusses MOFs as structures; emphasizes their synthesis strategies, properties, and applications; explores the existing methods of using immobilization processes of various enzymes; and lists their possible chemical modifications and combinations with other compounds to formulate the ideal supports for a given application.

## 1. Introduction

Enzymes have been widely used as natural biocatalysts in the pharmaceutical, chemical, and food industries, in addition to their well-known applications in medicine and in effluent and solid-waste treatment systems [1,2,3,4,5,6,7,8]. This is mainly due to the diversity of reactions enabled by biocatalysts, as well as their high efficiency, specificity, and selectivity [9,10,11,12,13,14,15,16,17,18]. Furthermore, enzymes are biocompatible and biodegradable structures that can be derived from renewable resources [19,20,21,22,23,24,25,26,27,28,29]. Unlike conventional organic syntheses, in enzymatic biocatalysis, the reactions of multifunctional molecules are carried out without the need for previous activation or the use of temporary protection for functional groups, resulting in more economical processes and less waste generation [29,30,31,32,33,34].

However, there are clear hurdles to the use of free enzymes, such as degradation (or denaturation) at high temperatures, the need for strict pH control during reactions, their difficult recovery and reuse, high production costs, and their instability under unfavorable environmental conditions, all of which hinder a more widespread implementation across different industries [19,20,22,30,31,35,36,37,38,39]. A suitable approach used for overcoming these problems is the immobilization of enzymes onto insoluble or solid supports [40,41,42,43,44,45,46,47,48,49,50]. Making them insoluble improves their operational characteristics under adverse conditions, which, in turn, enables their employment in media under more extreme temperatures, under comprehensive pH ranges, and in the presence of organic solvents instead of water [50]. Immobilization also allows for higher product quality and lower processing costs [51].

Another benefit to immobilization is the more efficient handling of enzymes through solid matrices in comparison to liquid-phase counterparts, which facilitates the separation of final products and reduces their contamination [52]. Additionally, immobilized enzymes show very little to no allergenicity, high recoveries, and a reuse capacity, rendering processes more economical [29,53]. To increase the stability of the enzymes during storage and make them more resistant to operational conditions, several types of support for enzyme immobilization have been studied, including magnetic nanoparticles, sol-gels, mesoporous silica, and polymers [[17],[27],[43],[54],[55],[56],[57],[58],[59],[60],[61],[62],[63],[64],[65],[66],[67],[68],[69],[70],[71],[72]. Immobilization technologies can also prevent subunit dissociation, aggregation, autolysis, and proteolysis, apart from delivering more suitable reaction microenvironments [73,74,75,76].

However, some challenges have also been observed in these techniques, such as low loading efficiency and enzyme denaturation due to incompatible incorporation processes [77]. In addition, conventional supports can present irregular non-uniform structures, which can impair the activity of the immobilized enzymes [78,79,80]. Among the several materials that can be used as supports for immobilized enzymes, metal–organic frameworks (MOFs) can be highlighted. These are an emerging class of porous materials built from the self-assembly of certain organic ligands and metal ions or specific clusters [81,82,83,84]. Their use as immobilization supports has been encouraged due to their inherent unique properties, such as structural flexibility, adjustable pore size, large surface area, and the possibility of post-synthetic modifications, among others [85]. The scientific relevance of using MOFs as support for enzymes can be observed by the significant increase in the number of published articles on these materials (Figure 1).

Furthermore, as observed in Figure 1, it is possible to discuss possibilities not yet evaluated for MOF applications as enzyme supports. As the figure presents and compares the number of MOF-related publications over the past 10 years, it is clear to see that the research on the topic is being carried out at an increasing pace, and it is possible to identify a vast field of present and future possibilities.

MOFs’ flexible structure size and porous environment, as well as their network of binding and interaction sites, allow for the immobilization of most enzymes and facilitates the mass transfer of substrates and products [86]. These materials have the highest surface areas ever reported for this specific application and can deliver high immobilization efficiencies due to their vast number of functional sites and pores [87,88]. Furthermore, MOFs can behave similarly to enzymes due to their inherent catalytic groups [86]. It is important to highlight, however, that for an efficient immobilization to occur, the enzyme confinement method must be carefully chosen, as any structural modification of the enzymes can lead to a significant reduction in their catalytic activity. Moreover, in case the interaction between the enzyme and the MOF is weak, enzyme leaching may occur. Therefore, it is essential to also evaluate the governing support–enzyme interactions [83].

Several authors have published reviews that discuss biocatalysts composed of enzymes and MOFs, addressing the most common methods of synthesis and immobilization of these composites, as well as their several applications [86,89,90,91,92,93,94,95,96,97,98,99]. In this scenario, this review intends to update the discussion of MOFs as highly relevant materials for a wide range of applications, as well as to discuss their roles and mechanisms of action as supports for enzymatic immobilization, and the different combinations for the formation of enzyme–MOF composites to diverse ends, such as catalysis, medicine, and in biosensor manufacturing, among others.

## 2. Metal–Organic Structures: Synthesis Strategies

MOFs are classes of chemicals that contain metallic ions (or coordinated metals) and organic ligands in their structures [100,101,102]. Thus, a MOF can be distinguished by a coordinated network with organic ligands (that can be mono-, di-, or trivalent, or tetravalent) containing empty spaces, or ‘pores’ [100,103,104]. The metal–ligand chemical bonds present in the composition of MOFs are predominantly of covalent nature and of the Lewis acid–base type (metal ion and ligand, in that order), given that they can generate a coordination composition [103,104]. Thus, the choice of metal and ligands ultimately determines the structure and pore size in MOFs.

There are no MOFs readily available in nature, except for stepanovite and zhemchuzhnikovite minerals [105]. Thus, the low functional stability of these materials in natural environments with characteristics of high crystallinity, microporosity (partially), high permanent surface area, and low thermal and chemical stability, in addition to their porosity and density, can substantiate both the interest in this field and the need for further studies on the use of MOFs in different areas [103,105,106,107].

As mentioned above, the unique characteristics of these materials combined with their wide range of applications reinforce the need for the development and improvement of synthesis techniques [105,106,107]. Currently, MOFs can be synthesized by different strategies, such as reticular synthesis [108], hydrothermal (solvothermal) routes [109,110,111], diffusion [112,113,114], electrochemistry [111,115,116], microwaves [117,118], mechanochemistry [119,120], heating, and ultrasound [111]. Figure 2 shows a schematic representation of the main strategies for MOF synthesis.

### 2.1. Methods of Synthesis

#### 2.1.1. Reticular Synthesis

Professor Omar Yaghi et al. [121] developed a synthesis strategy based on modular chemistry, known as reticular synthesis [121]. In this methodology, polytopic organic molecules bind to transition metal ions [121,122]. Subsequently, secondary building units (SBUs) are covalently linked across the entirety of the crystal [121,122,123]. SBUs are complexes or clusters in which ligand coordination modes and metal coordination geometries can be employed to modify these fragments into extended porous networks using polytopic structures [121,122,123].

In a later work, Yaghi et al. [108] discuss that reticular chemistry refers to the arrangement of pre-established coordinate structures through rigid molecular building blocks (the SBUs) that replicate and remain united through metal–ligand bonds [108]. Furthermore, in the same work, the authors redefined the term SBU, which was initially used to characterize fragments of zeolites, but was then defined around the geometry of the units classified as extension points [108].

In this way, for the construction of a broad network, SBUs must be structured in the correct mining, as this structuring guarantees the three-dimensionality of the material, so the geometry of the binder can directly influence the structure of the material [108,121].

#### 2.1.2. Hydrothermal–Solvothermal Synthesis

Hydrothermal synthesis was initially used for the production of zeolites. Later, it was also incorporated in the synthesis of MOFs [111]. Jarrah and Farhadi (2019) used hydrothermal synthesis to synthesize a MIL-101(Cr) and P2W18@MIL-101(Cr) nanohybrid. The nanohybrid was used in an adsorption test with the following organic dyes: methylene blue, rhodamine B, and methyl orange. The results indicated that the material obtained showed fast selective adsorption for systems with different dye concentrations [124].

In this technique, soluble impellers are used in a reactor, where the system operates under high pressures and temperatures (Figure 2a) [111,125]. The hydrothermal and solvothermal methods employed are dependent on the solvent used. In general, processes that use water as a solvent are termed hydrothermal processes [124], while those that use other solvents are classified as solvothermal processes [126].

The main advantages of this method reside in the good control of the morphology and the composition of the MOF [111]. It is worth mentioning that the rate of cooling can influence the properties of the synthesized material [109]. The main disadvantages of these methods include the processing time and the operating costs, making it difficult to reproduce them on industrial scales [111,124,125,126].

#### 2.1.3. Diffusion Synthesis

The synthesis method via diffusion is based on the gradual transport of several interacting species [112]. Diffusion-based methods of synthesis can be subdivided into two strategies [111]. In the first strategy, liquid solvent diffusion is performed [125]. First, two layers are formed at different density levels. The precipitating solvent resides in one of these layers, and then the final product in the solvent sits in the other [111]. This way, through the contact between the two interfaces, the gradual diffusion of the precipitant between the separated layer takes place, thus facilitating the crystal development (Figure 2b) [127].

In the second strategy, gradual diffusion occurs through physical barriers [128]. In addition, gels can also be used as an environment for diffusion and crystallization. This material is used because it mitigates the slow rate of diffusion and hinders the sole precipitation of MOFs [111,127,128].

#### 2.1.4. Electrochemical Synthesis

Electrochemical synthesis is widely used on an industrial scale to produce MOFs [129]. This methodology is based on principles of green chemistry since, when compared to the solvothermal method, for example, it imparts low costs, operating at lower pressures and temperatures, and requires shorter synthesis times, while presenting higher selectivity [130]. It is worth mentioning that during the crystallization step, issues may occur due to the development of metal ions in situ near the surface of the support, which reduces the agglomeration of crystals [131,132]. Figure 2c shows a schematic representation of an electrochemical synthesis of MOFs.

As with the hydrothermal method, the cracking process is thermally induced during temperature decay. However, as mentioned above, electrochemical synthesis occurs at milder temperatures, as compared to the former technique. According to Mueller and co-workers [111], the less abrupt cooling may favor the process of MOF formation [111,130,131].

The main disadvantage of this synthesis method compared to the hydrothermal route is the need for controlling a more significant number of variables, since parameters such as voltage and pulse, for example, need to be carefully adjusted [130,131,133].

#### 2.1.5. Microwave-Assisted Synthesis

The microwave-assisted technique is widely used for synthesizing small particles of oxides and metals [134]. Chen et al. [135], for example, performed the synthesis of MOF-74(Ni) with different methods, such as hydrothermal and microwave-assisted methods. The researchers evaluated the performance of these materials in the adsorption of CO_2_/N_2_ and verified that the MOF-74(Ni) synthesized by microwaves presented better adsorption performance. In addition, the authors reported that the protocol studied proved to be easy to conduct, and was also faster when compared to the other methods studied [135].

Through this process, it is also possible to increase the temperature of the solution, thus facilitating the formation of nanometric metal crystals [134,136]. It is worth mentioning that this strategy apparently cannot be directly used to synthesize MOF crystals [136]. However, it can speed up the synthesis process and adequately control the size and shape of MOFs [137]. Figure 2d presents a schematic representation of the use of microwaves in MOF synthesis.

Another aspect that needs to be considered is the control of parameters for solvent evaporation. Since temperature expansion can increase the solubility of crystals in saturated solutions, the process facilitates the formation of crystals during the cooling phase [134,136,137].

#### 2.1.6. Mechanochemical Synthesis

The mechanophysical strategy employs mechanical forces as a precursor of chemical reactions (Figure 2e). In this type of synthesis, chemical transformation is preceded by the mechanical rupture of intermolecular bonds [138,139]. Synthetic chemistry has employed mechanical activation in multicomponent reactions (ternary and higher) to form co-crystals with applications in the fields of pharmacy, organic synthesis, inorganic solid-state chemistry, and polymer science, among others [140].

Thus, several reasons are highlighted for using this strategy in the synthesis of MOFs. The main advantage of this method is the redced possible environmental impacts caused by the process. Syntheses in the absence of organic solvents can be carried out at room temperature, for example. Another positive aspect is reduced synthesis time [112,138,139,140].

#### 2.1.7. Sonochemical Synthesis

This methodology uses frequencies between 10 MHz and 20 KHz, which are higher than those detectable by the human ear (Figure 2f) [125]. The synthesis media can be close to a solid consistency if the cavitation and the microjets emitted during the reactions have the capacity for deterioration, activation, and interface variation [141], as well as for dispersion and agglomeration [142]. Alternatively, a liquid acts under pressure, specific temperature, and homogeneous conditions, or it is the interface that acts under the pressure of the medium, in case of forcing [141].

The main advantages of using sonochemical synthesis are the speed of synthesis, energy efficiency, process simplicity, and room-temperature reaction environments [111,138,141,142]. Yu et al. [143] employed the sonochemical route in the synthesis of Zn-based porphyrins MOF-525 and MOF-545. The authors obtained both porphyrins at high purity, and processing times were of 2.5 h and 0.5 h, respectively. It is worth noting that the materials showed excellent results also in the hydrolysis of dimethyl-4-nitrophenyl phosphate (DMNP) and in the adsorption of bisphenol-A (BPA), when compared to samples obtained conventionally [143]. Table 1 lists some methods of MOFs synthesis and their characteristics.

## 3. Metal–Organic Frameworks

### 3.1. Properties

Metal–organic frameworks (MOFs) have plated several roles in many industries and have become promising materials in the areas of catalysis, drug delivery, sensors, biological markers, pesticides, and others [153]. Their wide application is linked to their key physical properties and versatility, which are evidenced by organic structures linked to a central ion and, more specifically, a metallic cation [125]. The coordination sphere has a well-defined geometry, leading to the creation of crystals originating from this spatial arrangement, allowing pores to form in a polymerized manner. A scheme of the above definition is shown in Figure 3 [84].

Metal–organic frameworks show a wide variety of physicochemical and biological properties due to the versatility of their compositions (Figure 4) [154]. The binding of a metal ion or cluster to a flexible chain of organic polymers creates excellent magnetic properties in these composites that can be widely explored. This also facilitates the removal of these nanomaterials from their respective reaction media [99]. MOFs are also excellent precursors of chemical synthesis, depending on the chemical groups present in their organic part, where they act as activators or inhibitors of reaction points [155]. Additionally, they can act as electron donors or acceptors due to the properties of these structures being associated with coordination polymers, which behave as Lewis acids [156]. Many of these structures can interact with ionic or organic membranes and selectively migrate carrying ligands or macromolecules in biological media from one region to another [107]. The semiconduction properties of these materials also enable their use in the development of cutting-edge nanotechnology materials and processes. Owing to their excellent thermal capacity, new devices that require high sensitivity, easy detection and mapping, and good thermal stability can also be produced based on these inherent characteristics [157].

The ability of MOFs to act as catalysts or supports for the immobilization of biocatalysts renders these composites widely employable in chemical syntheses [158]. The chirality of these structures also enables favorable interactions with optically active materials, allowing for enhanced selectivity for these materials when in biological media [159]. Their thermal capacity, based on the metallic components, enables MOFs to integrate structures that require rapid cooling or heating [160]. Their semiconduction properties are also associated with the metallic center or the semiconducting organic ligands of these polymers, which allows for their use in nanotechnological applications [161].

Additionally, high porosity is one of the properties that add the most value to these materials, as pore sizes can be adjusted at the time of synthesis, according to the method and chemical precursors used [162]. The pores on the contact surface can act as housings to small molecules to be carried in fluids and organisms, and even to other organic molecules responsible for a given specific activity [163]. Luminescence, another key property, is characterized by the emission of light from the excited compounds. In MOFs, this is not only associated with the type of metal present in their composition but can be potentialized by organic ligands that present ideal chromophores for this property, such as aromatic structures [164,165].

### 3.2. Applications

#### 3.2.1. Adsorption

Adsorption is a fairly easy and low-cost technique that has been widely used, among other ends, to remove aquatic contaminants (Figure 5) [162]. MOFs are materials that can be successfully used in this technique due to their good adsorbent properties. More specifically, they have been employed in the removal of excess biological compounds, antibiotics, pesticides, gases, and other toxic pollutants, such as heavy metals [163]. Pan et al. [166], and Ghanem et al. [167] reported the adsorption process of organophosphate compounds used as herbicides, glufosinate (GLUF), glyphosate (GLY), and bialaphos (BIA) via MOFs. When metabolized, these compounds form derivatives that are frequently found in underground water bodies and in the soil, and that cause several environmental problems. They are also difficult to remove due to their high solubility and polarity. The adsorption process described made use of the magnetic properties of these MOFs, their high structural porosity, available surface area, and the possibility of compounds being quickly bound to the metallic center [165]. Thus, this becomes a viable technique, both from an environmental perspective and from an economic point of view, since MOFs can be reused for many cycles.

Antibiotics are drugs used to treat human and animal infections and have become an emerging environmental problem due to their excessive and incorrect disposal [168]. These compounds can be removed from aquatic systems using the MOF adsorption method, as reported in [169]. In addition to aiding the elimination of the aquatic contamination, these materials could also be used to remove polluting gases from the atmosphere via gas adsorption [170]. Many other materials are already widely used for this purpose, such as activated carbon and zeolites. However, they have shown a reduced ability to adsorb carbon dioxide [171]. Thus, materials made from metal–organic frameworks are highly promising, given their properties of adjustable pore size, easy handling and application, reuse, and selectivity [172]. In recent years, this versatility has led to a great interest in MOF, resulting in the use of these materials for different purposes. When associated with simple techniques, such as adsorption, many new options can be enabled.

#### 3.2.2. Catalysis

There has always been high demand for cheaper and faster processes in several industries. Therefore, the use of catalysts is widely studied for the optimization of industrial processes. MOFs, for example, can be used as catalysts for chemical reactions [173]. Given the aforementioned properties, they can provide high selectivity of substrates, and can be easily separated from reaction media and vastly reused (Figure 6). In the literature, several types of chemical reactions at small and large scales have been catalyzed by MOFs, including conventional catalysis [174,175], biocatalysis [173,176,177,178], and electrocatalysis [174,179]. The development and employment of these materials at industrial scales are significant, as they are excellent catalysts. However, it is still necessary to address the stability of MOFs under various reaction conditions, such as pH, temperature, and organic solvents, which has currently been a challenge for researchers.

#### 3.2.3. Drug Delivery

The number of biomedical applications of structures based on MOFs has been growing throughout the years due to the excellent versatility of these materials, high porosity, and large available surface area [132]. One of these key applications is in drug loading, which allows MOFs to work as carriers of the active compounds of various drugs through the body, from small organic molecules to macromolecules, such as nucleic acids and proteins (Figure 7) [180]. One issue related to this application is the toxicity of MOFs and the materials’ lack of full biocompatibility with the organism [181]. One advantage is that, due to their high loading capacity, they can be monitored in the body, allowing for the mapping of the reaction mechanism of different drugs, especially in the development of new drugs.

#### 3.2.4. Sensors

Biosensors are promising tools which can detect quick, selective, and sensitive molecules [182]. Due to the insulating characteristics of MOFs, they show great potential in the preparation of electrochemical sensors supported by carbon, which extends their application to the detection of analytes in different industrial fields, including environmental and biomedical fields, among others [183,184,185]. MOFs are great detectors of pollutants due to their affinity for specific groups of organic molecules [186]. Organic solvents, aromatic compounds, and heavy metal ions can also be detected using MOFs made from lanthanides [153,187].

Due to their adjustable pore size and high surface area, MOFs can also provide an ideal environment to accommodate analytes, allowing them to selectively absorb and release specific substrates through size recognition, effectively increasing signal and detection capabilities [188,189]. In addition, features such as the presence of metal coordination sites and lattice structures make them superior materials for the production of electrode coatings and for analyte detection [189]. Furthermore, there is the possibility of promoting the enhancement of their sensitivity to certain analytes through functionalization by immobilizing functional sites, initiating specific coordination, or promoting hydrogen bonding interactions with the target analyte [188].

MOF composites, formed by the incorporation of active biomolecules, such as antibodies, enzymes, and nucleic acids, can improve the selectivity, sensitivity, and detection limits of electrochemical sensors [190,191]. Biomolecule–MOF composites have been designed with an innovative focus on the detection of compounds of interest depending on the application sector. Some key compounds include uric acid [192,193], glucose [194], microRNAs [195], H_2_O_2_ [196], carcinoembryonic antigens [197], acetaminophen, and dopamine [193]. The main biomacromolecules are enzymes, as they provide more ecological, economical, and sustainable processes [29].

Enzymes can be incorporated into the structure of metal–organic structures and lead to the formation of sensitive electrochemiluminescence biosensors [88,198]. Examples include the manufacture of structures responsible for the detection of oncoproteins related to tumor proliferation (Figure 8), MOF enzymes of environmental interest [199], and other applications of industrial interest (such as the immobilization of enzymes for biocatalysis and the monitoring of biochemical reactions) [200].

Wang et al. [201] developed an enzymatic sensor for the photoelectrochemical detection of hypoxanthine using a nanoscale porphyrin MOF (Al-TCPP(Zn)) modified with the xanthine oxidase enzyme. Al-TCPP(Zn) exhibited an O_2_-dependent cathodic photocurrent, and this signal could be used for photoelectrochemical detection. After the addition of hypoxanthine, the produced biosensor delivered better responses due to the photoreduction of the H_2_O_2_ product catalyzed by xanthine oxidase. For the photoelectrochemical detection of hypoxanthine, the proposed sensor exhibited low detection limits, which was comparable to, or even better than, previous methods in terms of linear range and limits of determination; the selectivity was tested against several interferences, showing to have only been slightly affected. The authors also pointed out the reusability of the biosensor.

In Wang et al. [202], a glucose sensor for cascade biocatalysis constructed via the double confinement of enzymes in a nanocage-based zeolite imidazole (NC-ZIF) structure was evaluated. The enzyme@NC-ZIF showed good mass transport rates and excellent enzyme conformational versatility, due to the increased mesoporosity of the structure. The produced GOx/Hemin@NC-ZIF achieved good efficiency in catalytic cascade reactions in colorimetric and electrochemical glucose biosensors, enabling long-term quantitative analysis and continuous real-time monitoring of glucose in transpiration. Although the GOx/Hemin@NC-ZIF is very promising as a sensor, the method is limited to sweat tests, requiring further studies in order for other body fluids to be applied in innovative physiological and clinical investigations.

#### 3.2.5. Hydrogen Storage

MOFs can store hydrogen due to the large available surface of these materials [203]. Their hybrid metallic and molecular composition allows for several adjustments, such as the functionalization of possible ligands and their storage under variable temperatures [204]. MOFs have also become very promising in replacing noble metals during hydrogenation syntheses as Pt, the most commonly used metal to this end, is expensive and, even when compared to MOFs, shows lower yields in hydrogen trapping [205]. Therefore, a straightforward application of these hybrid nanomaterials is indicated, as they possess pores that serve as “gas pockets”, holding hydrogen atoms for synthesis (Figure 9).

#### 3.2.6. Environmental Applications

The environmental applications of metal–organic structures have been widely explored in recent years, as the growing drive to minimize the impacts of chemical residues has become the focus of extensive research around the world [206]. MOFs are used as efficient removers of heavy metals in fluids and aquatic environments [191]. They have been used to remove harmful gases and pollutants [207], such as carbon dioxide [208], based on their adsorption capacity [209]. Ma and colleagues [210] synthesized a MOF compound given its application as a biosensor of organophosphate pesticides, i.e., common pollutants in the agro-industry. These nanomaterials played a substantial role in the detection and removal of organic substances and solvents [211], organic dyes [212], antibiotics [213], volatile organic compounds [210] and other contaminants of industrial effluents [214].

Another essential environmental application is the detection of ammonia levels as a result of bioaccumulation, which has drawn the attention of environmentalists. Depending on concentration ranges, this can cause serious problems in aquatic food chains [215]. Thus, metal–organic structures are an excellent alternative for identifying levels of environmental pollutants [216] and in the treatment of effluents [217]. Their easy synthesis and high reuse rates render them particularly more accessible and targeted in the environmental area, which can be noted by the increase in the number of works published in recent years on this application [218].

All the applications discussed in this work present several possibilities of exploration in the industrial sector (Table 2). The flexible topology of these materials enables new architectures and, consequently, new properties and applications for MOFs, in addition to those that already exist and are extensively studied.

Thus, it is clear that nanomaterials have been widely used in different areas, which reinforces the need to develop, synthesize, and apply MOFs. A disadvantage of their use is still the high associated costs, with processes becoming economically unfeasible depending on their chemical composition, compared to other conventional structures. However, these nanoparticles are still very promising because such costs can potentially be counterbalanced by the number of possible reuses, the ease of synthesis, the wide range of applications, and the highly flexible structure for different processes. This is reinforced by a series of previously discussed properties, and those not yet tested in association with these materials, bringing the growing use of MOFs in complex industrial processes that benefit from the advancement of nanotechnology into perspective.

## 4. Enzyme Immobilization with Metal–Organic Frameworks (MOFs)

The immobilization of enzymes onto nanomaterials has revolutionized the use of these macromolecules in various industrial fields, which have been more recently enhanced by the advent of metal–organic frameworks [247]. The efficient immobilization of enzymes, i.e., its support and methods, is the result of perfect matching of factors depending on the enzyme [248]. Furthermore, the choice and success of the immobilization methods in the reaction depends of the different properties of the substrates and products, as well as the diversified applications of the products obtained. In addition, all methods have advantages and limitations. Consequently, the optimal immobilization conditions for a given enzyme are determined using experimental assays.

In addition to the main factors mentioned that influence the immobilization process, other parameters are important, such as pH, temperature, ionic strength, charge, and porosity of the support. These factors have a lesser or greater effect depending on the immobilization method. As previously mentioned, MOF characteristics of structural versatility, such as the porosity, large surface area, and organic–inorganic hybridity organization, render MOFs excellent candidates for enzyme immobilization using the most diverse methods (Figure 10) [93,99,247,249,250,251]. Regarding the porosity of the support, the mesoporous MOFs have been designed and constructed to obtain a high enzyme loading capacity and to reduce the diffusion resistance of reactants and products during the reaction. According to Xia et al. [93], the size of the pore openings may allow MOFs to gain size selectivity.

In Subtopics 4.1–4.4, immobilization studies using MOFs with different methods are presented.

### 4.1. In Situ Synthesis

In this method, the enzymes of interest and MOF materials (metal ions and organic ligands) are mixed under mild operating conditions in a suitable solution [93]. Using this immobilization technique, Wu, Yang, and Ge [252] assessed the stability behavior of some enzymes in organic solvents and compared these results with those obtained with the same proteins in their free form. To this end, lipase B from *Candida antarctica*, horseradish peroxidase, and cytochrome C were immobilized on the composite ZIF-8. The results showed that, even though the enzymes had different properties, the three immobilized macromolecules showed far superior stabilities in dimethyl formaldehyde, dimethyl sulfoxide, ethanol, and methanol compared to their free counterparts. Furthermore, the immobilized enzymes preserved almost 100% of their initial activity after incubation in the organic solvent, showing that the immobilization strategy protected them against potential denaturation due to the solvents used.

Another study considering MOF parameters was performed by Gascón et al. [253]. The researchers studied the synthesis and in situ strategies used to immobilize beta-glucosidase and laccase in nanocrystalline MOF platforms which aim to increase the activity of the tested enzymes. According to the results obtained, the immobilization stages in MOF nanocrystals favored the efficiency and the specific activity of the enzymes. Derivatives formed from in situ strategies showed an enzymatic charge above 85% and a loss of enzymatic activity of around 5%. Furthermore, the studied immobilization methodology effectively preserved the enzyme activity in a non-aqueous medium (N, N-dimethylformamide). Therefore, the researchers concluded that enzymes can be effectively immobilized in MOF nanocrystals and that in situ immobilization is a viable alternative in the preparation of immobilized biocatalysts.

Even though the in situ approach to immobilizing enzymes in MOF was efficiently conducted and requires mild reaction conditions, not all MOFs are ideal for this process. This is because the mode of enzyme dispersion and their subsequent location on the support can negatively affect the immobilization reactions [252].

### 4.2. Covalent Bonding

Unlike the in situ strategy, immobilization by covalent bonding occurs when the already-synthesized MOF is coated with substances capable of binding to the amino groups on the enzyme surface [254]. Many MOFs are susceptible to modification with functional groups to turn them into immobilization matrices [93].

Using this strategy, Cao and collaborators [255] immobilized soy epoxide hydrolase in UiO-66-NH2 MOF with glutaraldehyde as a binding agent, later applying this derivative in the biosynthesis of a (R)-1, 2-octanediol enantiomer. The results showed that the derivative presented a remarkable enzymatic load (87.3 mg/g), and recovered activity of 88%, as well as operational stabilities related to pH, temperature, and contact with organic solvents comparable to the frozen form of the enzyme under study. In addition to the improvements in the enzymatic characteristics associated with immobilization, the protein, when tested for the synthesis of (R)-1, 2-octanediol, delivered an enantiomeric excess of 81.2%. Therefore, the authors concluded that the immobilization of soy epoxide hydrolase on MOFs via covalent bonding showed strong potential for both improving enzyme characteristics and for being applied in enantiomeric reactions.

While seeking to further optimize the preparation and reuse of enzymes immobilized in MOFs, Wang et al. [251] incorporated iron oxide during MOF synthesis and used the final support to immobilize a *Candida rugosa* lipase via covalent bonding. The methodology employed by the researchers is justified by the ease of separating the derivative from a given reaction medium with the aid of a simple magnetic field. The derivative obtained was tested for the hydrolysis of olive oil and delivered a conversion rate of more than 65% after 6 h of reaction at 65 °C. Furthermore, the enzyme immobilized in the composite retained about 60% of its initial activity after 10 consecutive reaction cycles. Therefore, according to the above article, the synthesized support had both a large surface area and strong magnetic characteristics, which render this specific composite a good candidate support for enzyme immobilization.

### 4.3. Surface Immobilization

Surface immobilization (or adsorption) is the most widely used immobilization technique [254] due to the relatively low associated costs and the easy-to-perform methodology [93]. Because it is a versatile process, adsorption can be used to immobilize different enzymes on different supports, including MOFs [93,254,256,257]. In this technique, enzymes bind to the support through weak interactions such as van der Waals forces, hydrogen bonds, or electrostatic forces; therefore, they can be easily removed from the support via variations in pH and temperature, for example [93]. However, physical adsorption is still widely used and investigated due to its simplicity and the non-requirement for complex reagents [93,254].

In an attempt to compare advantages and disadvantages of this technique, Cao et al. [257] immobilized a lipase from *Bacillus subtilis* in a Cu-BTC-based MOF via physical adsorption and used the obtained derivative in an esterification reaction. The researchers obtained excellent results and demonstrated that the derivatives showed high operational stability and good enzymatic activity. Even after 10 consecutive reaction cycles, the lipase retained 90.7% of its initial activity and 99.6% of its initial conversion.

Another study on surface immobilization was performed by Pang and co-workers [256]. The researchers studied the support synthesis and the subsequent laccase immobilization on mesoporous Zr-MOF. According to the results, the laccase@Zr-MOF complex exhibited an adsorption capacity of 221.83 mg/g, wide temperature and pH distributions, and better stability when compared to that of the free laccase. In addition, the immobilized enzyme was able to maintain about 50% of its activity after 10 reaction cycles of contact between the derivative and ABTS, and retained 55.4% of its initial activity after three weeks of storage. With these numbers, the authors concluded that the immobilization method was successfully employed and that the synthesized support is a potential candidate for laccase immobilization via physical adsorption.

### 4.4. Entrapment

The immobilization strategy using entrapment or encapsulation is based on the confinement of the enzyme to a microenvironment located inside the support [93]. Contrary to other techniques, immobilization by entrapment causes isolation of the enzyme from the reaction medium, and also gives the protein better stability against potential denaturation caused by organic solvents, high temperatures, or sudden changes in pH [93,173]. Furthermore, using a MOF as support for this type of immobilization has extra advantages compared to other matrices: (i) MOFs can be synthesized according to their most suitable pore size (supports can have specific sizes for each type of substrate to allow for the efficient insertion and binding of the immobilized enzyme, reducing diffusional limitations); (ii) large enzyme loads can be achieved using MOFs as a consequence of their pore size; and (iii) encapsulated enzymes show a lower tendency to detach from the support [173].

Making use of such advantages, Li et al. [258] encapsulated a nerve agent detoxifying enzyme (organophosphorus acid hydrolase) in a mesoporous zirconium–MOF composite. The researchers reported that the synthesized support exhibited high enzyme loading capacity (12 wt%) and considerably improved thermal and storage stabilities.

In another study, Lian and co-workers [259] immobilized two enzymes in a tandem nanoreactor using a hierarchically structured MOF (PCN-888). The immobilized enzymes were glucose oxidase (GOx) and horseradish peroxidase (HRP). For the immobilization of both proteins to be successful, the researchers had to follow an encapsulation order: GOx followed by HRP. In the described process, the largest pores of the MOF (6.2 nm) were used to accommodate glucose oxidase, the 5.0 nm cavities accommodated horseradish peroxidase, and the smallest cavities (2.0 nm) remained unobstructed and accessible for the input of substrates and the output of products. Therefore, from the results, it was possible to conclude that the MOF was able to protect both enzymes against potential denaturation and considerably increased their operational stabilities (Table 3).

## 5. Future Trends

The application of MOFs combined with biocatalytic agents, including natural enzymes, is relatively recent. This integration has demonstrated an interesting synergistic performance in biocatalysis, due to the increased stability and reusability of encapsulated biocatalysts and the expansion of their applications into other fields [86,260]. Since the porosity properties of MOFs were identified, their investigation has developed exponentially [83]. However, although significant progress has been made, the investigation on enzyme–MOF composites is still in early stages, with many challenges still being a hurdle to the expansion of their applications [260]. The performance of this composites is influenced by several factors, including conformation; biomolecule activity and size; morphology; and the structural irregularity of particles in the design, preparation, and analysis of functionalized MOFs [82,86].

The use of MOFs for enzyme encapsulation is a fast developing field, and a significant increase in the number of studies on their properties in a short period of time leads us to believe that new highly effective biocatalysts are on the verge of being developed [261]. Great efforts have been made to this end; however, addressing the existing obstacles and improving current strategies are necessary so that enzyme–MOF composites can be fully suitable for practical applications [86,262]. There are expectations of future investigations in this area [260]. Challenges include the low diversity of biocompatible organic ligands and the toxicity of metals, in addition to the potential application of metals and ligands that have not yet been employed to this end [261,263].

To meet enzyme requirements of high activity and stability for practical applications and to elucidate the catalytic behavior of enzyme–MOF systems, it is necessary to investigate and improve the spatial structure of enzymes in MOFs [260,262]. This includes the establishment of spatial distributions that allow the confinement of multiple enzymes in MOFs, since the effective control over the location and orientation of enzymes can contribute to an increase in catalytic efficiency and a reduction in the resistance to the mass transfer of reagents [262]. In addition, exploring the suitable pore size and distribution profiles of MOFs is certainly an essential step in the encapsulation of several enzymes. Appropriate pore sizes can be optimized to meet specific criteria of enzyme accommodation, improving catalytic properties [260].

This review aimed to gather and discuss key information on MOFs, such as their synthesis, properties, and roles in enzyme encapsulation. We believe that the discussions, methodologies, and case studies presented can be helpful to readers and researchers interested in this topic. We also believe that this work can be used as a tool in the development of MOF-based materials for diverse applications, especially those related to enzymatic biocatalysis.

## 6. Conclusions

This review systematically reported on the mechanisms of action, latest advances, challenges, and future perspectives of the use of MOFs as support substrates in enzyme immobilization. MOFs are considered excellent candidates to support immobilization routes. This is because they present a wide variety of physicochemical and biological properties owing to the versatility of their composition. These impart properties include structural flexibility, adjustable pore size, large surface area, and the possibility of post-synthetic modifications, among others.

The chemistry of MOFs has developed exponentially since the porosity properties of these materials were identified. However, progress still needs to be made regarding the stability of MOFs under different reaction conditions (such as pH, temperature, and organic solvents), and in the storage of this material, constituting the most challenging aspects of their research. The elucidation of the different interactions between the MOF ‘housing’ and the enzymes that reside in their microenvironments during the various encapsulation processes is also paramount, since this can guide the construction of enzyme-MOF composites of high stability and bioactivity.

As the design and synthesis of MOFs with specific functionality at predetermined pore locations improve, interactions with biomolecules become more specific, resulting in more selective structures. Additionally, the recent methodologies and technologies based on computational chemistry can contribute to the development of new versatile projects of enzyme–MOF composites of high efficiency. However, to scale up laboratory-scale processes to larger scales, a more comprehensive understanding of the nature of enzyme–MOF composites is still required.

According to the discussion presented in this article, it can be concluded that enzymes immobilized on MOF supports clearly show better catalytic activity and operational stability than when compared to those obtained with their free form. In addition, such composites show an excellent maintenance of their initial activity after incubation in organic solvents by reaching a maximum percentage, which confirms that the immobilization strategies protect these proteins against possible solvent-related denaturation. Finally, it is expected that this review article, having presented synthesis strategies, properties, and applications of both MOFs and enzyme–MOF composites, can be a significant contribution to the advancement of the research on supports for enzymatic catalysis.

## Figures and Tables

**Figure 1 molecules-27-04529-f001:**
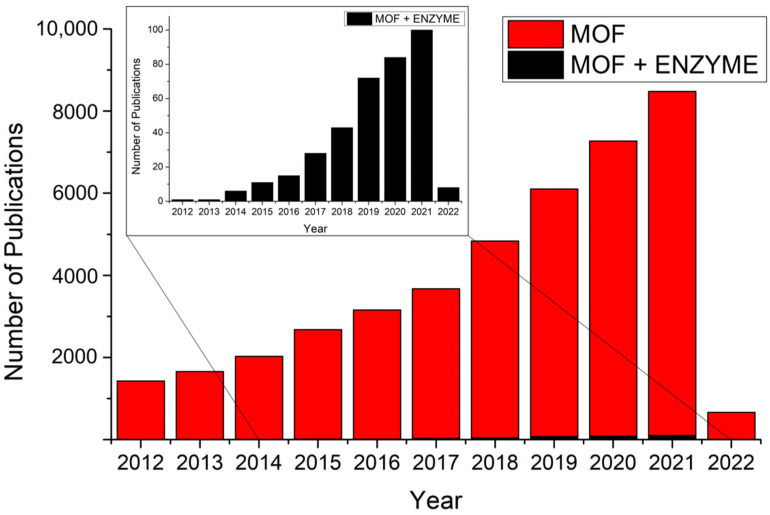
Growth in the number of published articles retrieved on Scopus over 10 years using the following keywords: (1) “metal-organic frameworks” and “MOF”; and (2) “metal-organic frameworks”, “MOF”, “enzyme”, and “immobilization”. The search was carried out on 7 January 2022, and returned (1) 45,925 and (2) 371 documents.

**Figure 2 molecules-27-04529-f002:**
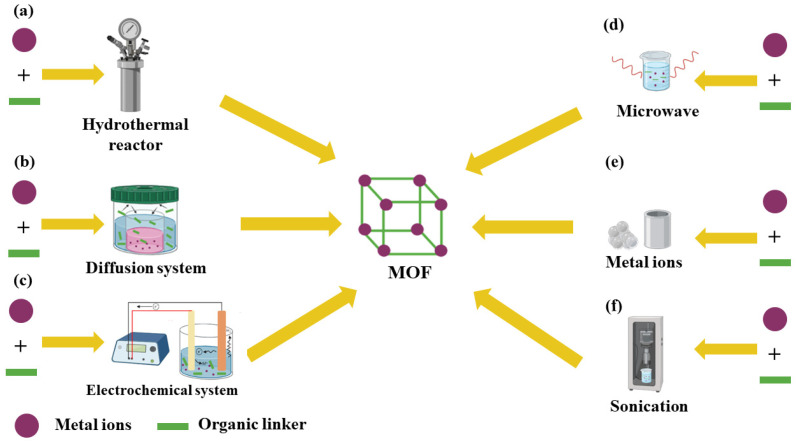
Schematic representation of strategies of (**a**) hydrothermal–solvothermal synthesis, (**b**) diffusion synthesis, (**c**) electrochemical synthesis, (**d**) microwave-assisted synthesis, (**e**) mechanochemical synthesis, and (**f**) sonochemistry synthesis.

**Figure 3 molecules-27-04529-f003:**
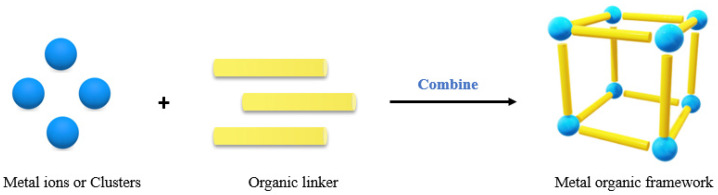
Representation of the formation of a crystalline structure of metal–organic structures (MOFs) based on organic ligands being coupled to a metallic center.

**Figure 4 molecules-27-04529-f004:**
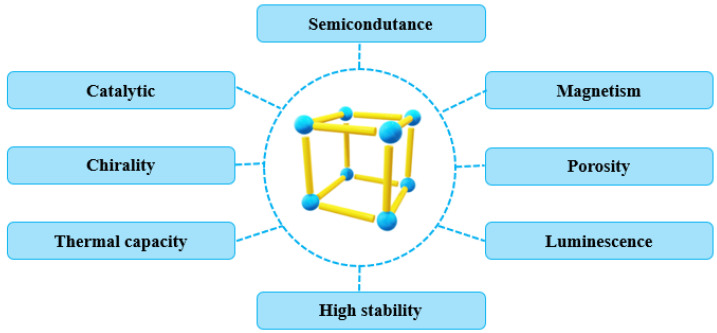
Representation scheme of composite properties based on metal–organic framework (MOF) structures, highlighting their thermal capacity, chirality, high stability, semiconductivity, luminescence, magnetism, catalytic power, and porosity.

**Figure 5 molecules-27-04529-f005:**
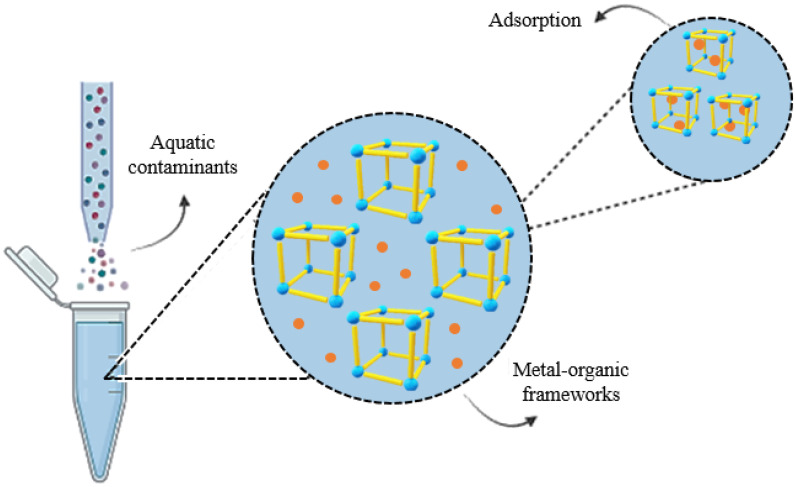
Schematic diagram showing pollutant adsorption on the surface of metal–organic frameworks (MOFs), where the contaminant particles can bind to the material, leaving a pollutant-free aqueous medium.

**Figure 6 molecules-27-04529-f006:**
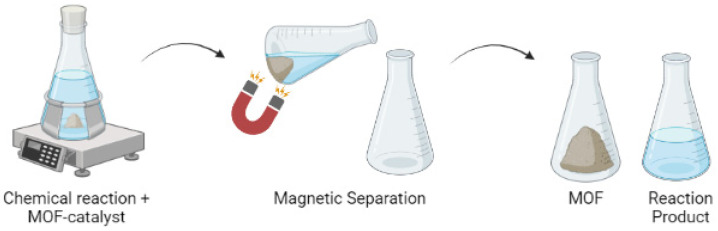
A simplified representation of the separation of metal–organic structures (MOFs) from their reaction media by their magnetic properties, which enables their simplified removal—an excellent characteristic for catalysts.

**Figure 7 molecules-27-04529-f007:**
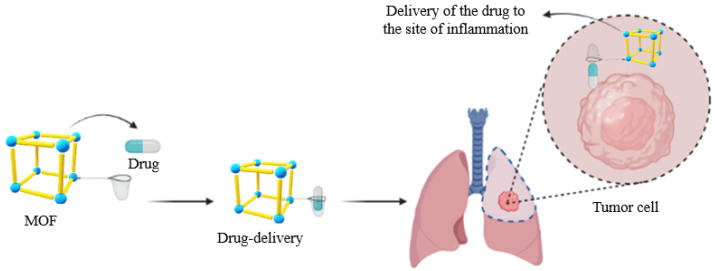
Representation of a metal–organic framework (MOF) as a drug-administrating carrier in tumor cells. They can be used as identifiers of the regions of inflammation and, due to their luminescence, can make it easy to detect the exact region of drug action.

**Figure 8 molecules-27-04529-f008:**
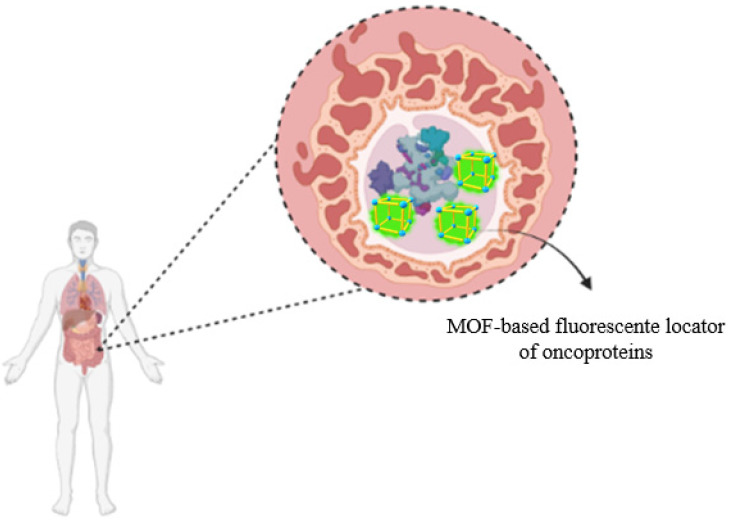
Illustration of the mapping of overexpressed macromolecules in tumor cell lines through luminescent metal–organic framework (MOF) composites. This is a widely explored property, which was enabled by their metallic centers and mapped by confocal microscopy.

**Figure 9 molecules-27-04529-f009:**
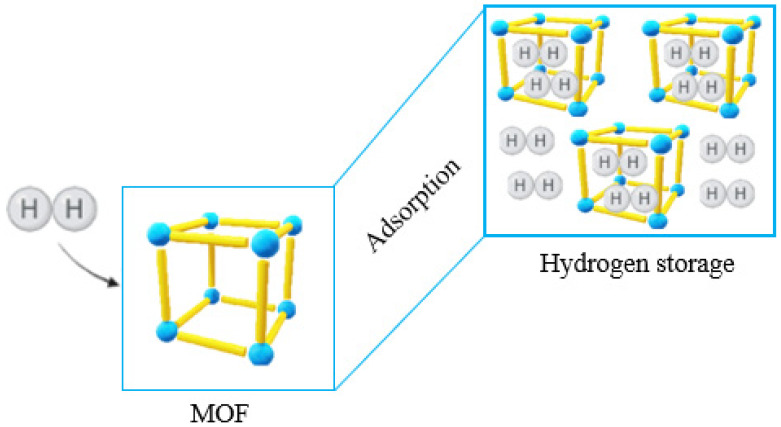
Scheme of metal–organic frameworks (MOFs) capturing hydrogen via their adsorbent properties, a function that can be used for hydrogen storage.

**Figure 10 molecules-27-04529-f010:**
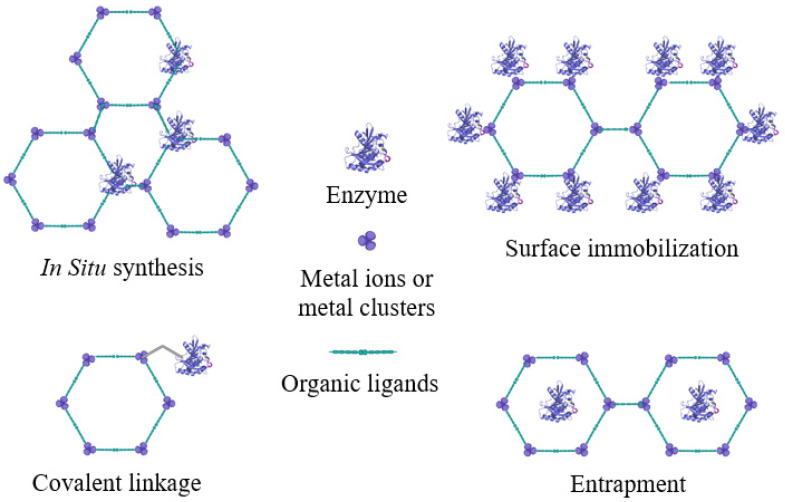
Representation of different techniques of enzyme immobilization onto MOFs.

**Table 1 molecules-27-04529-t001:** Metal–organic frameworks: a summary of different synthesis strategies and their applications.

Synthesis Strategy	Main Features	Applications	Material	Ref.
Hydrothermal (Solvothermal)	Generally, processes that use water as a solvent are termed hydrothermal processes, while those that use other solvents are classified as solvothermal processes [124,126]	Dye removal	MIL-101(Fe)@PDopa@Fe_3_O_4_	[144]
		Lithium–sulfur battery	Cu_2_(CuTCPP)	[145]
Diffusion	Diffusion MOF synthesis methods can be subdivided into two strategies: diffusion between two liquids with different densities (no physical barrier) and gradual diffusion that occurs through physical barriers [111]	Drug delivery	CD-MOF	[146]
		Adsorption of copper ions	MOF-5	[112]
Electrochemical	Electrochemical synthesis is widely used at an industrial scale to produce MOFs [129]	Lithium-ion batteries	Zn-POMCF	[129]
		Ibuprofen adsorption	[Zn(1,3-bdc)0.5(bzim)]	[147]
Microwave-assisted	The microwave technique is widely used in synthesizing small particles of oxides and metals [137]	Gas separation	MOF-74	[148]
		CO_2_ capture	MOF-5	[149]
Mechanochemical	In this type of synthesis, chemical transformation is preceded by the mechanical rupture of intermolecular bonds [139,140]	Drug delivery	Cu-MOF/IBU@GM	[150]
		Drug delivery	ZIF-8@alginate NPs	[151]
Sonochemistry	This methodology uses frequencies between 10 MHz and 20 KHz, which are higher than those detectable by the human ear, for dispersion and agglomeration purposes [102]	Adsorption of antibiotics	[Zn6(IDC)4(OH)2(Hprz)2]n	[152]
		DMNP hydrolysis and BPA adsorption	MOF-525 and MOF-545	[143]

**Table 2 molecules-27-04529-t002:** General applications of composite metal–organic frameworks (MOFs) reported in the scientific literature, and their main areas of interest, such as environmental and biomedical industries, among others.

N_0_	MOFs	Enzyme	Applications	Ref.
1	ZIF-90/Ce-MOF	Catalase	Sensitive detection and degradation of hydrogen peroxide	[219,220]
2	L-MOFs	Glucose oxidase	Insulin delivery	[85,221]
3	PCN-333(Fe)	Alcohol Dehydrogenase	Catalysis of the conversion of toxic levels of alcohols to aldehydes in cells	[181,222]
4	MIL-101(Cr)	Microperoxidase 8	Dual catalytic activity in the selective oxidation of organic molecules	[180,223,224]
5	AgNC/Mo(II)-NS	Cholesterol oxidase	Detection and concentration in blood vessels or other body tissues	[225,226]
6	QDs/CDs@MOFs	Ascorbate oxidase	Improved ascorbic acid detection	[227,228]
7	ZIF-8	Lactate/glucose oxidase	Tumor cell mapping and energy reduction for tumor cycles	[229]
8	UiO-66	Lipase	Drug synthesis against venous thromboembolism	[230,231]
9	OMUiO-66 (Ce)	Glutamate oxidase	Screening of specific chiral amino acids in complex biological samples	[198,232]
10	ZIF-8	Glucose oxidase	Electrochemical glucose detection	[186]
11	MIL-88B-NH2(Cr)	Trypsin	Protein degradation by enzymatic hydrolysis	[99,233]
12	ZIF-8	Glucose oxidase	Electrochemical glucose detection	[99]
13	Tb-mesoMOF	Mb	Oxidation of ABTS and THB	[99,234]
14	ZIF-8	Urease	Sensitive biosensor for urea detection	[235]
15	CYCU-4	Trypsin	Protein digestion	[99,236]
16	HKUST-1	Peroxidase	CO_2_ adsorption	[99,213,237]
17	UIO66-NH2	Acetylcholinesterase	Biosensors for organophosphorus pesticide detection	[166,210]
18	MOF-199	Laccase	Removal of heavy metals from fluids and aquatic environments	[238,239]
19	QD-MOF	Oxidase	Degradation of organic dyes in industrial wastewaters	[240,241,242]
20	L-MOFs	Lipase	Luminescent sensors for environmental pollutants	[125,243]
21	ZIF-90	Catalase	Effluent treatment in wastewater	[214,216,244]
22	ZIF-67	Glucose oxidase	Antimicrobial agent	[244,245]
23	Ce (III)/UiO-66	Hydrolases	Adsorptive removal of organic dyes from aqueous solutions	[214,216]
24	ZIF-8	Choline oxidase	Detection and removal of water pollutants	[215,246]

**Table 3 molecules-27-04529-t003:** Advantages and disadvantages of different enzyme immobilization strategies in/onto MOF.

Strategy	Advantages	Disadvantages	Ref.
In situ synthesis	Easily conducted; requires only mild reaction conditions	Not all MOFs are ideal candidates to the process	[252]
Covalent bonding	The enzyme is strongly attached to the surface of the support; several MOFs can be used	It can change the morphology of the enzyme, altering its activity or even inactivating it	[93,251]
Surface immobilization (adsorption)	Relative low cost and simple methodology	Enzymes can be easily leached from supports due to variations in pH and temperature	[93,254]
Entrapment	Gives proteins greater stability against denaturation caused by organic solvents, high temperatures, or sudden changes in pH	Mass transfer limitations may occur; difficult for substrates to reach the active site of enzymes	[93,173]

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
