# Peer review of "The Chemistry and Applications of Metal–Organic Frameworks (MOFs) as Industrial Enzyme Immobilization Systems"

_molecules, 2022, doi:10.3390/molecules27144529_

Round 1
Reviewer 1 Report
In this paper, authors discuss MOFs as structures, emphasizing their synthesis strategies, properties and applications, exploring the existing methods of using in the immobilization processes of various enzymes, and listing their possible chemical modifications and combinations with other compounds to formulate the ideal support for a given application. It is good review, but some need to be attention:
(1) Line 243, “processing times were of 2.5h and 0.5h” should a space between number and “h”.
(2) Line 307-308, Please check “[109,110] re-ported the adsorption process of organophosphate compounds used as herbicides, glufosinate (GLUF), glyphosate (GLY), and bialaphos (BIA) via MOFs.”
(3) Please check the format for strain name.
(4) It is better to list methods of synthesis in a table for their characteristic.
(5) It is better to supply discussion of some factors for immobilization processes of enzymes.
Author Response
Dear Editors,
We would like to thank the reviewers for their crucial and constructive criticism, which significantly improved our work's scientific quality. Please find below the answers to their comments and questions. For ease of reference, all modifications were marked in yellow in the revised manuscript version.
Reviewer
Review Report (1)
Comments and Suggestions for Authors
In this paper, authors discuss MOFs as structures, emphasizing their synthesis strategies, properties and applications, exploring the existing methods of using in the immobilization processes of various enzymes, and listing their possible chemical modifications and combinations with other compounds to formulate the ideal support for a given application. It is good review, but some need to be attention:
Dear reviewer, we are very grateful for the recommended suggestions. We believe that they were very important for the improvement of our manuscript.
(1) Line 243, “processing times were of 2.5h and 0.5h” should a space between number and “h”.
Response: Suggestion accepted. Please be advised that we have changed the text and left it highlighted for your verification.
(2) Line 307-308, Please check “[109,110] reported the adsorption process of organophosphate compounds used as herbicides, glufosinate (GLUF), glyphosate (GLY), and bialaphos (BIA) via MOFs.”
Response: We appreciate the correction. We have checked the sentence and the references number have been corrected. New sentence in the manuscript: “…reported the adsorption process of organophosphate compounds used as herbicides, glufosinate (GLUF), glyphosate (GLY), and bialaphos (BIA) via MOFs.
(3) Please check the format for strain name.
Response: The uggestion was accepted. The strain names were revised and has been corrected.
(4) It is better to list methods of synthesis in a table for their characteristic.
Response: We appreciate the sugestion. New table (Table 1) was added in the manuscript with the methods of synthesis and their characteristics. New sentence in the manuscript: “The Table 1 lists some methods of MOFs synthesis and their characteristics.”
(5) It is better to supply discussion of some factors for immobilization processes of enzymes.
Response: The suggestion was accepted. The authors added in the manuscript a brief discussion of some factors for immobilization processes of enzymes. As the discussion of immobilization parameters is something very broad due to they are influenced by the enzyme, the support and the method of immobilization, we cited the most important. However, we present the studies of the enzyme immobilization using the MOFs as support, mentioning which parameters the reported authors evaluated.
New sentence in the manuscript: “Efficient immobilization methods are the result of perfect matching of factors depending on the enzyme, of the support and methods of the immobilization. Also, the choice and success of the immobilization methods in the reaction depends of the different properties of the substrates and products and the diversified applications of the products obtained. In addition, all methods have advantages and limitations. Consequently, the optimal immobilization conditions for a given enzyme are determined by experimental assays.
In addition to the main factors mentioned that influence the immobilization process, other parameters are important, such as pH, temperature, ionic strength, charge, and porosity of the support. These factors have a lesser or greater effect depending on the immobilization method. Regarding the porosity of the support, the mesoporous MOFs have been designed and constructed to obtain a high enzyme loading capacity and to reduce the diffusion resistance of reactants and products during the reaction. According to Xia et al., the size of the pore openings may allow MOFs to gain size selectivity.
In subtopics 4.1 to 4.4 are presented immobilization studies using MOFs by different methods.
Reviewer 2 Report
The reviewed paper by J.C.S dos Santos et al. presented the applications of MOFs as matrices for the industrial immobilization of enzymes. The topic is hot, and recently it has been reviewed many times.
Remarks:
1. Nearly half of the abstract's text can be considered a general introduction. An expected reader is for sure familiar with the MOF definition etc.
2. Please, make your fig. 1 more readable; just turn it into a bar graph.
3. The manuscript is comprehensible. However, it doesn't strictly refer to the title: no truly industrial examples (but one), no patent cited.
4. My inspection of the Web of Science base for <MOF & enzyme & industrial> revealed ten reviews for the last five years, and only two are cited. The cited references are generally relevant; nevertheless, important previous reviews seem missing. For the omitted, see, for instance: Perez, V.G.; Sanchez-Sanchez, M., Environmentally Friendly Enzyme Immobilization on MOF Materials in Immobilization of Enzymes and Cells: Methods and Protocols in Molecular Biology, vol. 2100, 4th Edition, New York, N.Y., 2020, pp.271-296; doi.org/10.1007/978-1-0716-0215-7_18.
Author Response
Dear Editors,
We would like to thank the reviewers for their crucial and constructive criticism, which significantly improved our work's scientific quality. Please find below the answers to their comments and questions. For ease of reference, all modifications were marked in yellow in the revised manuscript version.
Review Report (2)
Comments and Suggestions for Authors
The reviewed paper by J.C.S dos Santos et al. presented the applications of MOFs as
matrices for the industrial immobilization of enzymes. The topic is hot, and recently it
has been reviewed many times.
Dear reviewer, we are very grateful for the recommended suggestions. We believe that they were very important for the improvement of our manuscript.
Remarks:
- Nearly half of the abstract's text can be considered a general introduction. An
expected reader is for sure familiar with the MOF definition etc.
Response: Thank you for your comments and contributions. All suggestions were incorporated into the new version of the manuscript. Changes are in the revised manuscript.
- Please, make your fig. 1 more readable; just turn it into a bar graph.
Response: Thank you for your comments and contributions. All suggestions were incorporated into the new version of the manuscript. Changes are in the revised manuscript.
- The manuscript is comprehensible. However, it doesn't strictly refer to the title: no truly industrial examples (but one), no patent cited.
Response: Thank you for your comments and contributions. We believe that the methods presented are bold and innovative, so we chose not to change the manuscript's title.
- My inspection of the Web of Science base for <MOF & enzyme & industrial> revealed ten reviews for the last five years, and only two are cited. The cited references are generally relevant; nevertheless, important previous reviews seem missing. For the omitted, see, for instance: Perez, V.G.; Sanchez-Sanchez, M., Environmentally Friendly Enzyme Immobilization on MOF Materials in Immobilization of Enzymes and Cells:
Methods and Protocols in Molecular Biology, vol. 2100, 4th Edition, New York, N.Y.,
2020, pp.271-296; doi.org/10.1007/978-1-0716-0215-7_18.
Response: Thank you for your comments and contributions. All suggestions were incorporated into the new version of the manuscript. Changes are in the revised manuscript.
Reviewer 3 Report
In this review authors summarized the properties, applications and synthesis of MOFs and focused their importance as supports for enzyme immobilization. The article is well designed and pleasantly to read. And the content is sufficient to highlight the issues and summarize the global efforts on the issue. Also, the readability of the posted images is conducive to understanding the content being read. In my opinion, the article is suitable for publication as submitted, however I believe authors could tried to address the following comments:
Comment 1: The first 16th references could be updated.
Comment 2: One summarized table could be prepared to cover the Methods of Synthesis (2.1.).
Comment 3: The choice of the word “industrial” in the title could be considered a little bold.
Author Response
Dear Editors,
We would like to thank the reviewers for their crucial and constructive criticism, which significantly improved our work's scientific quality. Please find below the answers to their comments and questions. For ease of reference, all modifications were marked in yellow in the revised manuscript version.
Review Report (3)
Comments and Suggestions for Authors
In this review authors summarized the properties, applications and synthesis of MOFs and focused their importance as supports for enzyme immobilization. The article is well
designed and pleasantly to read. And the content is sufficient to highlight the issues and
summarize the global efforts on the issue. Also, the readability of the posted images is conducive to understanding the content being read. In my opinion, the article is suitable
for publication as submitted, however I believe authors could tried to address the following comments:
Dear reviewer, we are very grateful for the recommended suggestions. We believe that they were very important for the improvement of our manuscript.
Comment 1: The first 16 th references could be updated.
Response: Thank you for your comments and contributions. All suggestions were incorporated into the new version of the manuscript. Changes are in the revised manuscript.
Comment 2: One summarized table could be prepared to cover the Methods of
Synthesis (2.1.).
Response: We appreciate the sugestion. New table (Table 1) was added in the manuscript with the methods of synthesis and their characteristics. New sentence in the manuscript: “The Table 1 lists some methods of MOFs synthesis and their characteristics.”
Comment 3: The choice of the word “industrial” in the title could be considered a little
bold.
Response: Thank you for your comments and contributions. We believe that the methods presented are bold and innovative, so we chose not to change the manuscript's title.
Round 2
Reviewer 2 Report
Still, in my opinion, the title doesn’t exactly correspond to the content. There are no real industrial applications.